# The Intersection of Food Security and Mental Health in the Pursuit of Sustainable Development Goals

**DOI:** 10.3390/nu16132036

**Published:** 2024-06-27

**Authors:** Helen Onyeaka, Ovinuchi Ejiohuo, Oluseyi Rotimi Taiwo, Nnabueze Darlington Nnaji, Omowale A. Odeyemi, Keru Duan, Ogueri Nwaiwu, Olumide Odeyemi

**Affiliations:** 1School of Chemical Engineering, University of Birmingham, Edgbaston, Birmingham B152TT, UK; n.nnaji@bham.ac.uk (N.D.N.); o.nwaiwu@bham.ac.uk (O.N.); 2Department of Psychiatric Genetics, Poznan University of Medical Sciences, 60-806 Poznan, Poland; 3Doctoral School, Poznan University of Medical Sciences, Bukowska 70, 60-812 Poznan, Poland; 4Molecular and Cell Biology Unit, Poznan University of Medical Sciences, 60-572 Poznan, Poland; 5Faculty of Veterinary Medicine, University of Ibadan, Ibadan 200132, Oyo, Nigeria; oluseyitaiwo7@gmail.com; 6Department of Microbiology, University of Nigeria, Nsukka, Enugu 410001, Enugu, Nigeria; 7College of Nursing, Obafemi Awolowo University Teaching Hospital Complex, Ile Ife 220005, Osun, Nigeria; oaodeyemi@gmail.com; 8Birmingham Business School, Department of Management, University of Birmingham, University House Edgbaston Park Road, Birmingham B15 2TY, UK; kxd226@student.bham.ac.uk; 9Ecology and Biodiversity Centre, Institute for Marine and Antarctic Studies (IMAS), University of Tasmania, Launceston, TAS 7004, Australia; olumide.odeyemi@utas.edu.au

**Keywords:** food insecurity, mental health, sustainable solutions, food security, mental well-being, sustainable agricultural practices, community food programs, traditional food banks, policy innovations, sustainable development

## Abstract

Food insecurity, a multifaceted global challenge, intertwines with mental health concerns, necessitating nuanced strategies for sustainable solutions. The intricate web of challenges posed by these intersections has made it imperative to delineate a strategic way forward, incorporating solutions and robust policy recommendations. This study aims to comprehensively examine the intricate relationship between food security and its intersection with mental health on a global scale, offering insights into case studies, responses, and innovative approaches to inform effective strategies for addressing these pressing challenges. This study involved an analysis of a literature search, mainly between 2013 and 2023, with an updated addition of relevant 2024 studies. Examining responses across regions unveils varied interventions, from targeted social safety net programs in West Africa to technology-driven solutions in Asia. Success stories, such as Ghana’s sustainable agricultural practices and Canada’s income transfer programs, underscore the efficacy of multifaceted approaches. Innovative initiatives like community food programs offer promising alternatives to traditional food banks. Furthermore, international cooperation and policy innovations, exemplified by the European Union’s “Farm to Fork Strategy”, demonstrate the potential for collective action in addressing food insecurity. By prioritizing integrated strategies, global collaboration, and evidence-based policymaking, we lay the groundwork for sustainable development where communities thrive nutritionally and mentally. We emphasize continuous research and evaluation and incorporating mental health support into community programs to pave the way for a future where communities are not only food-secure but also mentally resilient.

## 1. Introduction

Food insecurity is regarded as the unavailability of food required for well-being and affects mental health [1]. Where there is food insecurity, public health is greatly affected, which is why the United Nations included zero hunger in its Sustainable Development Goals (SDGs). The SDGs’ number two goal is to end hunger and achieve food security by 2030. Food security in our context is considered as access to safe and nutritious food [2] and the opposite of food insecurity defined above. Data from the UN [3] show that one in three people worldwide suffer moderate to severe food insecurity, translating to over two billion humans. Also, in that report, it was highlighted that more than 600 million people may face hunger globally by the year 2030 and that malnutrition lingers worldwide and jeopardizes the well-being and future of children under the age of five. In addition to this, recent data from the State of Food Insecurity Report for 2023 also show that about 29.6 percent of the global population—2.4 billion people—were moderately or severely food-insecure in 2022, of which about 900 million (11.3 percent of people in the world) were severely food-insecure.

There is an old saying that a hungry man is an angry man, and the data from the UN report [3] suggest that there are many angry people worldwide. The target year of 2030 set by the UN in its SDGs to achieve food security appears unattainable when their updated progress report [3] is considered. This suggests that millions worldwide will not have the desired nourished minds and bodies and good mental health in the future. Despite this gloomy outlook, some communities have experienced an improvement in their food security by implementing the recommendations of the UN [4]. The link between mental health and food security [4] or vice versa has not been properly established despite numerous studies; hence, it is very important to explore the nexus further, so that the factors that prevent good mental health due to food insecurity can be identified for further action.

A comprehensive survey [5] of up to 160 countries worldwide found that the well-being, food insecurity, and mental health of young persons were strongly related. The survey highlighted that, in less food-insecure environments, those facing chronic hunger and meal skipping are denied resources vital for mental health. Other investigators also found that deprivation of basic needs (food source, access, availability, and nutritional status) can affect mental health [6]. In Africa, where basic need deprivation is rife and prevalent in more than half of the population, a systematic review by Trudell et al. in 2021 [7] highlighted that the consequences of food insecurity on mental health are more prevalent among the elderly and women. Also, another study by Aguiar et al. in 2022, comprising mainly women (72% of 882) in Portugal, found that individuals from food-insecure households had more anxiety and depression symptoms [8]. Based on the observations so far, addressing food insecurity without dealing with mental health issues brings about a reductive intervention, and an integrative approach may be better. Reducing food scarcity can promote better mental health [4].

Studies of the effect of food insecurity on mental health using other population groups like the youth [5] and adolescents with a medical condition like diabetes [9] have also been investigated, and the mental predicament from worrying about food sufficiency was highlighted. Other investigations involving parents and children [10] have also been carried out elsewhere, and other reports elucidating the deleterious effect of food insecurity on mental health during uncertain situations like the COVID-19 pandemic [11] or being a refugee [12] have been shown.

This paper aims to comprehensively examine the intricate relationship between food insecurity and mental health globally, offering insights into case studies, responses, and innovative approaches to inform effective strategies for addressing these pressing challenges.

## 2. Methodology

This study employs a comprehensive and systematic methodology to investigate the intersection of food insecurity and mental health across various regions and socioeconomic contexts. The methodology comprises several key stages: a literature review, data collection, case study selection, and data analysis until May 2024. Each stage is detailed below to clarify the parameters and processes involved.

### 2.1. Literature Review

A thorough literature review was conducted to establish a foundational understanding of the existing research on food insecurity and mental health. The review focused on identifying key themes, trends, and gaps in the literature. The following databases were utilized:
PubMed: For peer-reviewed articles related to public health and epidemiology.Google Scholar: For a broad range of scholarly articles, including grey literature.Scopus: For comprehensive coverage of interdisciplinary studies.

Keywords and search terms included “food insecurity”, “mental health”, “social protection policies”, “nutritional interventions”, “case studies”, and “regional analysis”. Boolean operators (AND, OR, NOT) were used to refine searches and ensure relevant results. The criteria for the inclusion of articles were relevance to the intersection of food insecurity and mental health, publication in peer-reviewed journals, and empirical data or robust theoretical discussions. Exclusion criteria included articles that did not directly address the connection between food insecurity and mental health or lacked methodological rigor. The literature search initially covered articles published from 2013 to 2023 to capture recent developments and trends, with an updated analysis and inclusion of 2024 studies and literature searches up to May 2024.

### 2.2. Data Collection

Data collection involved gathering quantitative and qualitative data from multiple sources to gain a holistic view of the subject matter. The primary sources of data included the following:
Peer-Reviewed Journals: Articles providing empirical evidence and theoretical discussions.Government Reports: Documents from national and international bodies, such as the Food and Agriculture Organization (FAO) and the World Health Organization (WHO).NGO Reports: Publications from non-governmental organizations actively involved in addressing food insecurity and mental health issues.Case Studies: Detailed examinations of specific regions or programs implementing notable interventions.

To ensure journal quality, only peer-reviewed journals indexed in recognized databases such as PubMed and Scopus were included. Articles were assessed for methodological rigor, relevance, and impact.

### 2.3. Case Study Selection

Case studies were selected based on their relevance, diversity, and the availability of detailed documentation. The selection criteria included the following:
Geographical Diversity: Ensuring representation from different continents, including Africa, Europe, North America, and Asia.Intervention Types: Covering various approaches, such as social safety nets, technology-driven solutions, and community-based programs.Outcome Data: Availability of data on the impact of interventions on food insecurity and mental health outcomes.

Selected case studies include interventions in West and Central Africa, Somalia, Kenya, Nigeria, the European Union, the United States, Canada, and Asia. Each case study provided insights into different strategies’ effectiveness and contextual challenges.

## 3. Understanding Food Security

Almost a billion people worldwide lack access to sufficient, high-quality food in a manner that is socially acceptable for them to lead active lives, making food security a global concern [13,14].

According to the FAO, food security exists when the global human population always has physical and socioeconomic access to safe, adequate, and nutritious food capable of meeting their dietary needs and food preferences for an active and healthy lifestyle [14]. This robust definition of food security was crafted to include critical dimensions of food availability, accessibility, and utilization. Over time, other concepts were included, such as the stability and sustainability of food systems [15].

The term “food availability” refers to the steady and unbroken supply of enough quantity and quality—to support a healthy lifestyle [16]. Food availability also considers sociocultural acceptability and food safety, ensuring that food is provided in a manner that respects human dignity and is devoid of microbial or toxic substances that could endanger human health [16]. Food availability is a crucial aspect of food security and has often been the focus of many initiatives to increase global food production [17]. It also involves building efficient transportation systems to ensure that food is available even in remote areas from their sources [18].

Food availability is closely related to food access; however, availability does not necessarily predict access. Food access encompasses the measures and resources required to guarantee that an adequate quantity of safe, nourishing, and socially acceptable food is within an individual’s reach. These resources span various aspects, including economic status (finances) and sociocultural (indigenous food practices) and physical (dependable transportation networks) aspects [19]. Access as a social science concept also describes the food choices available to individuals based on their socioeconomic status, education, and geographical location [20]. Utilization relates to how individual and household knowledge, food preferences, and health status affect the consumption and absorption of a balanced diet capable of meeting all their nutritional dietary needs [20]. Utilization also encompasses food processing and storage practices, sanitation and hygiene, and child care (FANTA 2006). The concepts of stability and sustainability were included much later in the definition of food security [21]. Stability refers to the ability of food agrosystems to withstand man-made or natural shocks or disruptions and food’s availability, access, and utilization [22]. Sustainability describes the need for efforts and measures to ensure the long-term ability of food systems to provide adequate food while preserving the environment, natural resources, and agro-ecosystems [15,22].

After years of decline, there has been an increase in global hunger since 2014 [23]. Globally, the percentage of undernourished individuals rose to 10.6% in 2015 and, subsequently, 11% in 2016 [23]. The FAO estimates that 821 million people worldwide, or almost one in nine people, were undernourished in 2017 [23]. The 2018 food insecurity statistics by the FAO also report that up to 60% of hungry people worldwide are women, and up to 5 million children below the age of 5 die from malnutrition-related causes annually [24]. The rise in food insecurity suggests that the Sustainable Development Goal (SDG) aim of ending hunger by 2030 may not be met [24].

Food insecurity is most evident in developing countries, characterized by low per capita income [25]. A study by Jones in 2016, which analyzed cross-sectional data from the 2014 Gallup World Poll across 149 countries, determined the prevalence of severe food insecurity to be as high as 33.8% in Sub-Saharan Africa and the prevalence of any form (mild, moderate or severe) to be as high as 76.1% in Sub-Saharan Africa [26]. This was further validated by a 2017 study by the ERS suggesting that up to 31.7% of Sub-Saharan Africa’s population is food-insecure [27]. Forecasts also indicate that this figure will likely remain higher than 20% by 2027. Asia has the most significant number of food-insecure people. In Asia and the Pacific, a projected 460.2 million people suffered from extreme food insecurity in 2021—a rise of 170.6 million, or 58.9%, from 2014. In 2021, the region comprised 49.8% of the 923.7 million people worldwide. Compared to 28.0 million in Eastern Asia, 17.4 million in South-eastern Asia, and 2.0 million in Oceania, Southern Asia had 412.9 million people who were extremely food-insecure [24]. In the United States, a study in 2015 revealed up to a 12.7% prevalence of food insecurity, affecting almost 16 million households and impacting women, children, and racial minorities more [28,29].

## 4. The Psychological Impact of Hunger

Hunger refers to a state of stress occasioned by an involuntary absence of food and insufficient calories [30]. Hunger rates are the highest in Sub-Saharan Africa (Figure 1). Conversely, South Asia exhibits markedly higher prevalence rates than the Americas and East Asia [31]. Incidence rates within North America and Europe remain below 2.5%. According to The Global State of Food Security and Nutrition report, over 0.8 billion 820 people worldwide are hungry, with the majority of older people residing in Sub-Saharan Africa [32]. These are dire statistics considering the widespread ramifications of food insecurity and malnutrition on mental health. According to WHO, mental health is “a state of well-being in which the individual realizes his or her own abilities, can cope with the normal stresses of life, can work productively and fruitfully, and can contribute to his or her community” [32].

Even though food insecurity is primarily concerned with nutrition and diet and directly impacts body weight [33], it also has well-noted impacts on human psychological well-being [6]. Conditions of food insecurity encompass issues of psychological acceptability, leading to feelings of deprivation and anxiety over food supply. Consequently, food insecurity can negatively influence psychological health [34]. A global study by Jones analyzing the link between food insecurity and mental health across individuals in 149 countries demonstrated a strong dose–response relationship between poor mental health and food insecurity [26]. According to the study, the increasing severity of food insecurity can be linked with a steady decline in mental health [26]. A similar comprehensive study by Frongillo, which evaluated 138 countries in the 2014 Gallup World Poll for food insecurity and subjective well-being, reported worry, sadness, or stress among individuals living under conditions of food insecurity [35].

A review of twenty-seven qualitative and quantitative studies from developing nations by Weaver and Hadley reported that food insecurity is correlated with anxiety, shame, stress, resignation, and depression [30]. According to a systematic study by Weaver and Hadley in 2009, there is a bidirectional association between mental health and food insecurity in which hunger increases poor emotional health and vice versa [28]. Other psychological symptoms, such as suicidal ideation, anxiety, and sleep disorders, have also been linked to food insecurity and hunger [28].

Numerous studies have demonstrated the long-term effects of nutrition on children’s health and development [36]. Adequate nutrition is essential for the correct development of the fetal brain during prenatal and neonatal development, according to research on mental health and cognitive development [37]. Food insecurity, mainly due to its role in micronutrient deficiency, is correlated with early growth faltering, altered cognitive development, and poorer scores on the mental health indices [38]. Children in food-insecure households are also typically exposed to parental aggression and psychosocial stress, leading to negative child behaviors [14,26,37,39,40].

A large study by Grineski et al. in 2018 showed the impact of food insecurity on children’s academic competencies, executive functioning, and social skills [41]. The results highlighted the significant detrimental effects of food insecurity on mathematics, self-control, and working memory. Other diminished parameters in children from food-insecure households include executive function and interpersonal skills. According to the study, a slip into food insecurity was occasioned by lower math and working memory scores [41].

Studies have shown that children in food-insecure households are not only aware of their limited food resources but are physically and psychologically influenced by it [42]. Fram et al. (2011) conducted a study of children aged 9 to 16 years in South Carolina and revealed that children showed cognitive, emotional, and physical awareness of household food insecurity, with emotional awareness characterized by worry, sadness, and anger [43]. In another study conducted in the United Kingdom, children aged 5 to 11 years described the experiences of frequent hunger, hunger when food was unavailable, and hunger at bedtime [44]. A study of Indian teenage girls aged 13–19 by Rani et al. (2018) also found that adolescent girls from food-insecure households were more likely to have high levels of anxiety, depression, loss of behavioral control, and psychological distress compared with those living in food-secure households [45]. Furthermore, research by Fertig (2019) at the University of Kentucky Center for Poverty indicates that mental distress due to food insecurity may linger on even into adulthood [46].

In a United States study on adolescents, insufficient food was closely linked to both depressive disorders and suicidal symptoms based on national data analysis. The study also characterized recovery during refeeding by a slow but gradual elimination of the symptoms [13]. Research has also posited possible biological mechanisms by which food insufficiency is associated with a depressed mood. According to a study by Polivy, chronic dieters or individuals with restricted diets often tend to have heightened emotional responsiveness and increased irritability and distraction [47].

## 5. Linking Food Security to the Sustainable Development Goals

Food security is directly linked to Sustainable Development Goal 2 (zero hunger), which aims to end hunger by 2030. Achieving this goal requires adequate food availability through sustainable agriculture and ensuring access to food for all individuals. Food security is guaranteed when appropriate measures are implemented to achieve zero hunger. These include food stability and adequate food stock to meet food demand [48]. Having access to healthy food is required for healthy living. This aligns with Sustainable Development Goal Three (SDG 3), which aims at good health and well-being. Parents who cannot afford healthy food could be stressed mentally. This implies that food insecurity impacts the mental health of an individual. Depression, anxiety, and psychological distress are among the reported mental health consequences, underscoring the urgency of addressing this issue [49].

Research demonstrates the detrimental impact of food insecurity on mental health, with the significant prevalence of mental illness resulting from it elevating it to a public health issue demanding immediate intervention [50]. For example, in a recent study on the association of food security with mental health among youths in Canada by Men et al. in 2021, it was reported that severe food insecurity caused poor health among the participants irrespective of their socioeconomic background [50]. The mental health impact of food insecurity could be exacerbated by pandemics, droughts, and natural disasters such as earthquakes and floods [51]. The pandemic affected the food supply chain due to the closure of borders across the globe [52]. Many became worried about how to obtain food as no one knew how long the pandemic would last, and many individuals could not afford the prices of available food. Students, workers, and families were all impacted.

The World Food Conference in 1974 emphasized the importance of food for physical and mental development, igniting global awareness of food insecurity [53]. In the United States, an estimated 13.7 million people experience this challenge, while a devastating 675.1 million in Africa lack access to sufficient food [7]. Global food insecurity disproportionately affects vulnerable groups [10]. Food security is crucial for a sustainable future amidst global challenges like poverty, injustice, and climate change. Therefore, achieving food security is not an isolated goal but an intertwined stepping stone towards fulfilling the broader vision embodied in the UN Sustainable Development Goals (SDGs).

Aside from the fact that food security is vital towards achieving SDGs 2 and 3, food security is required to achieve other goals. For example, a student who has not eaten might not entirely concentrate in the classroom, thereby affecting the fulfillment of SDG 4, which is access to quality education. While the government or private sector could provide quality education, a hungry student may not be able to achieve it. Often, they may be absent from school, thereby impacting their learning. During the COVID-19 pandemic, the mental health of both domestic and international students in Australia was affected by food insecurity in a study reported in [54].

Farmers affected by drought could not afford quality education for their children [55]. While the drought impacted the mental health of the parents, the children’s mental health could also be impacted by not being able to learn like their other classmates. Due to drought affecting the availability and accessibility of food, the following Sustainable Development Goals will be affected in such families: SDG 1 (no poverty—the farmer cannot afford basic needs for their children), SDG 2 (zero hunger—the farmer cannot afford to provide food for their family), SDG 3 (good health and well-being—the farmer’s health and the well-being of the children are impacted), and SDG 4 (quality education—the farmer will not be able to send their children to school due to lack of funds).

## 6. Case Studies and Global Perspectives

Food insecurity, a complex and multifaceted challenge, remains a pressing global challenge, affecting populations across various countries and regions. Food security and mental health are intimately intertwined, with their intersection posing critical challenges and opportunities for sustainable development globally. In this section, we delve into case studies from diverse geographical areas, highlighting the impacts of food insecurity and the responses implemented to address this critical issue. Additionally, we explore success stories and innovative approaches that have shown promise in improving food security and mental well-being. Examining these case studies and global perspectives aims to contribute to a nuanced understanding of the multifaceted nature of food insecurity and inform future strategies for mitigation.

### 6.1. Impact of Food Insecurity

A critical aspect of understanding the global landscape of food insecurity involves examining its varied impacts on different regions. One striking example is the West and Central Africa region, which has grappled with an unprecedented food security crisis for two consecutive years. The compounded consequences of the 2021 cereal deficit, security challenges, and the COVID-19 pandemic have led to a cereal production deficit of approximately 9.5 million tons, significantly affecting the region’s availability and affordability of food [56]. Contrastingly, the European Union (EU) faces distinct challenges in the wake of the Russia–Ukraine conflict. The war has disrupted global supply chains, leading to shortages in critical agricultural inputs, including energy, animal feed, and fertilizers [57]. The impact on the agricultural sector is substantial, with the EU heavily dependent on Russian fertilizer imports [58]. Additionally, the export restrictions imposed by Russia on essential commodities have contributed to food price surges in the EU, affecting both availability and affordability [59]. A comparative analysis of these regions highlights the diverse nature of food security challenges. While West and Central Africa grapple with the immediate consequences of production deficits and inflation, the EU navigates disruptions in the supply chain and seeks to mitigate the impact of export restrictions. Understanding these distinct challenges is crucial for tailoring effective responses.

Studies have also shown the impact of food insecurity on mental health. A study by Pryor et al. (2016) provides insights from a community sample focusing on young adults [60]. The study underscores the risks posed by food insecurity, including an increased likelihood of suicidal ideation, depression, and substance use. Maynard et al. (2018) explore the link between food insecurity and mental health among females in high-income countries; the study emphasizes the role of toxic stress in intertwining food insecurity and mental health [61].

The study conducted by Elgar et al. (2021) in the United States showed a negative association between relative food insecurity and psychological functioning, even after controlling for food insecurity [5]. This finding suggests that individuals experiencing food insecurity not only struggle with the tangible challenges of accessing sufficient and nutritious food but also battle with the psychological impact of observing others who are less worried about securing their next meal. The study highlights the global patterns observed in the relationship between relative food insecurity and mental health outcomes. The salience of reference groups becomes evident, with associations between food insecurity and poorer mental health intensifying in settings where the problem is less prevalent. This underscores the role of social comparisons in shaping perceptions of well-being and life satisfaction. Interestingly, the study found a diminished association with relative deprivation in high-prevalence contexts, where food insecurity is more commonplace, and individuals may be less likely to experience the same degree of relative deprivation compared to those in low-prevalence settings.

### 6.2. Responses to Food Insecurity

Understanding the multifaceted response to food insecurity across various regions (Table 1) offers critical insights into the effectiveness of social protection policies and intervention strategies. Governments have implemented over 50 measures to address the food crisis in West and Central Africa. These measures include targeted social safety net programs, tax reductions on food items, the release of existing food stocks, export restrictions, and subsidies on agricultural inputs [56]. However, the effectiveness of these measures is currently questionable, as markets in many countries face high food prices and poor availability of essential commodities.

In countries like Somalia, Kenya, Nigeria, Ethiopia, and the Sahel region, the Red Cross has been actively addressing extreme hunger and food insecurity [62]. Ground teams provide comprehensive support, including water, food, immediate financial aid, nutrition services, and healthcare. The Somali Red Crescent, operating in Somalia, has supported over 500,000 people through clinics, mobile health services, and financial aid for essential items [62]. Similarly, the Kenyan Red Cross and the Nigerian Red Cross have reached over 520,000 and 98,000 people, respectively, providing vital support through food, clean drinking water, health services, and financial assistance [62]. While these interventions alleviate immediate hunger, it is crucial to recognize that long-term solutions must complement emergency relief efforts to build resilience within communities facing climate change and systemic challenges.

In the European Union, responses to food insecurity are characterized by a multifaceted approach. Recognizing the importance of international cooperation, the EU has regularly met with international and multilateral bodies, such as the FAO, the World Trade Organization (WTO), and the Group of Seven (G7). These collaborations aim to promote humanitarian interventions and address the challenges posed by disruptions in the supply chain [59].

Europe provides a broader perspective on the impact of social protection spending on food insecurity [63]. Social protection expenditures throughout Europe generally encompass the overall support level through initiatives offering financial assistance and non-monetary aid, such as housing allowances [64]. During the Great Recession, countries with higher per capita investment in social protection spending exhibited lower levels of food insecurity [32,63]. Loopstra (2018) considered the cumulative generosity of interventions, encompassing cash benefits and in-kind transfers, including housing subsidies. The buffering effect of spending on housing subsidies on the relationship between rising unemployment and food insecurity underscores the interconnectedness of social protection measures. While income transfers are vital, addressing housing affordability and providing universal healthcare are integral components of a comprehensive strategy to combat food insecurity.

The Supplemental Nutrition Assistance Program (SNAP) is a notable case study addressing food insecurity through social protection policies in the United States. SNAP is a means-tested entitlement program that financially supports eligible households who purchase essential foods [65]. While studies reveal a significant decline in food insecurity among participants [63,66,67], persistently high levels suggest support levels and administration adjustments. The paternalistic nature of SNAP raises questions about its impact on households’ ability to maximize utility, prompting a critical examination of social protection programs and the necessity for continuous evaluation and adaptation to address food insecurity effectively.

Canada’s Universal Childcare Benefit (UCCB) offers insights into the impact of income transfers on food insecurity. Initiated in 2006, the UCCB provides a monthly income supplement of CAD 100 for each child under 6 years old nationwide [68]. Evaluations from the implementation of the UCCB program showed that using a CAD 1200 UCCB for children under <6 years lowered food insecurity by 2.4 percentage points, highlighting the potential of targeted income transfers to positively influence household food security [69]. Studies by Li et al. (2016) and Loopstra (2018) exploring social assistance programs and poverty reduction interventions in Canada demonstrate significant reductions in food insecurity during increased investment in cash transfer programs, emphasizing the crucial role of income support in comprehensive social protection policies [63,70].

The emergence of food banks has become a prominent intervention in addressing food insecurity, notably in high-income countries [71]. Over recent years, the growing prevalence of food banks in the UK mirrors a broader trend observed in other affluent nations [72]. Food banks, also known as food pantries, serve as charitable institutions where individuals can access groceries free of charge [73]. Common features of food banks include reliance on donated food, either from the community or surpluses donated by the food industry, and dependence on volunteers for their operation. Despite their prevalence in high-income countries such as Canada, the United States, and Australia, food banks face inherent limitations in effectively addressing food insecurity [63]. Studies suggest that users of food banks often experience high levels of severe food insecurity, and the assistance provided may need to be improved to prevent households from facing hunger [74,75]. The nutritional inadequacy of foods distributed by food banks is a common concern, with reports indicating insufficient amounts of essential nutrients such as fruits, vegetables, calcium, vitamin A, and vitamin C [76].

The impact of food banks on food insecurity is complex, while some research, such as the study by Roncarolo et al. in 2015, claims a reduction in food insecurity attributed to food bank interventions and methodological issues, including the absence of true baseline measures and high attrition rates, which call for cautious interpretation [77]. Additionally, users may experience feelings of shame about resorting to food banks, contributing to the underutilization of these services [78]. Notably, only a fraction of the food-insecure population tends to access food banks, highlighting potential limitations in addressing the widespread issue [8].

In response to the limitations of traditional food banks, organizations have explored alternative approaches, often called community food programs [79]. A summary of responses by region is provided in Table 1. These initiatives aim to provide sustainable alternatives by offering additional services beyond essential food provision [80]. Examples include community food centers, community kitchens, and community shops. In the United Kingdom and Canada, some food banks operate as part of community food centers, where users can access services like benefits counseling, debt counseling, cooking classes, and community kitchens [81]. The effectiveness of community food programs in reducing household food insecurity remains to be determined due to challenges in study design, including the lack of randomized trials and long-term follow-ups.

### 6.3. Success Stories and Innovative Approaches

While challenges are evident, notable success stories and innovative approaches offer glimpses of hope amid the global food security crisis. One such success story emerges from countries in West Africa that have implemented targeted social safety net programs. These programs aim to provide direct support to vulnerable groups, including children, women, youth, and the elderly, helping them cope with the impact of rising food prices. By focusing on social protection, these countries are addressing the immediate needs of their populations [56].

Innovative approaches to improving food security are evident in the European Union’s “Farm to Fork Strategy” [82]. This long-term plan emphasizes innovations such as precision farming, new genomic techniques, improved nutrient management, and integrated pest management to boost agricultural productivity. Additionally, the EU actively pursues trade agreements and policies to reduce dependency on a few countries and tackle global issues like climate change. These strategies reflect a commitment to resilience building and sustainable agricultural practices [59].

Sub-Saharan Africa has faced persistent challenges related to food security, but success stories and innovative approaches have emerged. Ghana, for instance, has made significant strides in improving food security through sustainable agricultural practices and targeted policy interventions. The government’s Planting for Food and Jobs program, launched in 2017, has boosted agricultural productivity and enhanced food security [83]. The program has positively impacted the quantity and quality of available food by providing subsidized inputs, technical support, and market linkages to farmers. Such success stories underscore the importance of multifaceted interventions that address immediate food access, sustainable agricultural practices, and livelihood support.

Technology-driven solutions have played a pivotal role in improving food security in Asia, particularly in countries like India and China. Mobile applications and data analytics have been leveraged to enhance the efficiency of food distribution systems, reduce food wastage, and ensure timely access to nutritional information [84]. For example, India’s “e-PDS” (electronic Public Distribution System) integrates technology to streamline the distribution of subsidized food grains to eligible beneficiaries [85]. This improves the effectiveness of food distribution and facilitates better targeting of vulnerable populations. The intersection of technology and food security presents a promising avenue for scalable and sustainable interventions that address the diverse needs of populations across regions.

Some programs have adopted innovative approaches to improving food access in addressing food insecurity. Subsidized fruit and vegetable box programs, such as the Good Food Box program, aim to increase access to fresh produce [86]. Evaluations of such programs have shown mixed results, with enrolled individuals experiencing a lower prevalence of food insecurity, while discontinuation of the program was associated with increased food insecurity. The success of these innovative approaches is contingent on factors like program design, sustained participation, and addressing the diverse needs of vulnerable populations. In emphasizing context-specific interventions, the case studies underscore the need for policies that recognize the diverse sociocultural, economic, and environmental factors.

Additionally, success stories from Sub-Saharan Africa illustrate the effectiveness of holistic approaches that extend beyond immediate food access, incorporating sustainable agricultural practices, livelihood support, and market linkages to foster enduring improvements. Moreover, integrating technology-driven solutions in Asia showcases the potential for achieving scale and efficiency in interventions. The adoption of mobile applications, data analytics, and electronic distribution systems emerges as a strategic avenue to streamline processes, minimize resource wastage, and enhance overall effectiveness in addressing the evolving challenges associated with food security.

## 7. The Way Forward

The intricate web of challenges posed by food security and mental health intersections has made it imperative to delineate a strategic way forward, incorporating nuanced solutions and robust policy recommendations. Integrating efforts addressing both aspects is vital for fostering sustainable development and achieving the United Nations’ SDGs. Scholarly insights underscore the need for comprehensive, integrated strategies that acknowledge the symbiotic relationship between food security and mental health [4,28]. Rather than isolated interventions, a holistic framework encompassing healthcare, nutrition, and social support is essential. This demands collaborative efforts from public health institutions, governmental bodies, and non-governmental organizations.

Robust social protection policies have proven instrumental in mitigating the adverse effects of food insecurity on mental health. Countries with higher per capita investment in social protection spending exhibited lower levels of food insecurity during economic downturns [32,63]. Therefore, policymakers should prioritize investments in social protection programs, ensuring they encompass financial aid, housing allowances, and other non-monetary assistance to address the multifaceted nature of food insecurity [64].

International cooperation is paramount given the global nature of food insecurity and mental health challenges. Collaborative efforts between nations, facilitated by organizations such as the FAO, the WTO, and the G7, can bolster humanitarian interventions and address disruptions in the food supply chain [59]. Shared resources, knowledge exchange, and joint research initiatives can amplify the impact of interventions.

The “Farm to Fork Strategy” employed by the European Union demonstrates the potential of innovative approaches to enhancing food security [82]. Policymakers should prioritize investments in sustainable agriculture practices, leveraging technologies like precision farming and improved nutrient management to bolster agricultural productivity. By ensuring the long-term resilience of food systems, these approaches contribute to food security and the overall well-being of communities.

To break the cycle of the bidirectional association between food insecurity and mental health, interventions should explicitly address mental health concerns. Integrating mental health support into existing community programs, especially in regions with prevalent food insecurity, can provide a comprehensive approach [28,60]. This requires collaboration between mental health professionals, community leaders, and policymakers.

The dynamic nature of the challenges at the intersection of food security and mental health necessitates ongoing research and evaluation. Policymakers should allocate resources for studies exploring the effectiveness of interventions, identifying gaps, and adapting strategies based on evidence [75]. Continuous evaluation ensures that policies remain responsive to evolving socioeconomic conditions and emerging mental health concerns.

The way forward demands a strategic marriage of social, economic, and healthcare policies that recognize the interconnectedness of food security and mental health. The success of these endeavors relies on a commitment to integrated approaches, global collaboration, and a steadfast dedication to evidence-based policymaking. By adopting such a holistic perspective, we pave the way for a future where communities are food-secure and mentally resilient, laying the foundation for sustainable and equitable development.

## 8. Conclusions

The intricate examination of food insecurity and its intersection with mental health presented in this study underscores the urgency and complexity of addressing these global challenges. Comprehensive analysis of case studies from diverse regions, responses, and innovative approaches provides a nuanced understanding of the multifaceted nature of food security. Notably, the documented impact on mental health emphasizes the pressing need for integrated strategies encompassing healthcare, nutrition, and social support. The success stories from different parts of the world and innovative approaches offer glimpses of hope amid the crisis. The presented solutions, ranging from targeted social safety net programs to technology-driven interventions, highlight the adaptability required to address the diverse sociocultural, economic, and environmental factors contributing to food insecurity. As this study outlines, the way forward calls for a strategic marriage of social, economic, and healthcare policies committed to global collaboration and evidence-based policymaking. The emphasis on continuous research and evaluation, incorporating mental health support into community programs, and recognizing the interconnectedness of food security and mental health pave the way for a future where communities are not only food-secure but also mentally resilient. Adopting a holistic perspective lays the foundation for sustainable and equitable development, fostering a world where well-being and access to sufficient and nutritious food are prioritized.

The strength of our study lies in its examination of the complex relationship between food insecurity and mental health globally, emphasizing diverse case studies and innovative interventions. It underscores the need for integrated strategies and robust policy recommendations to address these intertwined challenges effectively. Key strengths include its methodology, extensive literature review from 2013 to 2024, and insightful case studies from various regions, highlighting successful approaches and the importance of holistic, evidence-based policymaking. We advocate for continuous research, international cooperation, and integrating mental health support into community programs to achieve sustainable development where nutritional and mental well-being are prioritized.

While this study on food insecurity and mental health presents valuable insights, several limitations exist. The study focuses mainly on case studies, responses from selected regions, and interventions. This may limit the generalizability of findings to other geographical and socioeconomic contexts. There might be inconsistencies or variations in how food insecurity and mental health outcomes are measured across different studies and regions, affecting the comparability and robustness of conclusions. The study may also face challenges in establishing causal relationships between interventions and outcomes due to the complexity of factors influencing food security and mental health. We also did not delve deeper into how these recommendations can be practically implemented and evaluated across diverse policy landscapes. However, this study provides a clear pathway for future research, including areas where additional data, methodologies, or policy experiments could further advance understanding and interventions to tackle the interconnected challenges of food insecurity and mental health on a global scale.

## Figures and Tables

**Figure 1 nutrients-16-02036-f001:**
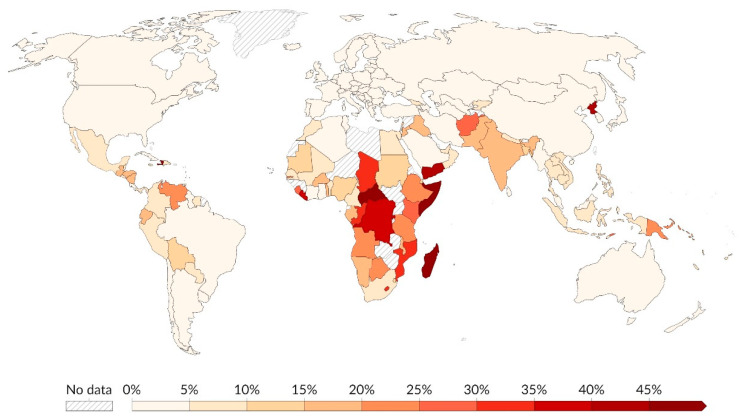
Share of the population that is undernourished (2001–2020) [31].

**Table 1 nutrients-16-02036-t001:** Summary of responses to food insecurity by region.

Region	Key Responses and Interventions
West and Central Africa	Targeted social safety net programs. Tax reductions on food items. Release of existing food stocks. Export restrictions—Subsidies on agricultural inputs.
Somalia, Kenya, Nigeria	Red Cross interventions providing water, food, financial aid, nutrition services, and healthcare. Support for over 500,000 people in Somalia through clinics, mobile health services, and financial aid.
European Union	Engagement with international bodies like FAO, WTO, and G7. Collaborative efforts to promote humanitarian interventions and address supply chain disruptions.
United States	Supplemental Nutrition Assistance Program (SNAP). Means-tested entitlement program providing financial support for purchasing essential foods.
Canada	Universal Childcare Benefit (UCCB). Monthly income supplement for children under 6 years old—reduction in food insecurity by 2.4 percentage points.
High-income countries	Emergence of food banks, limited effectiveness due to nutritional inadequacy and underutilization.
Ghana	Success stories in Ghana with sustainable agriculture and policy interventions.
Asia	Technology-driven solutions like India’s e-PDS system.

## Data Availability

Not applicable.

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
