# Peer review of "The Intersection of Food Security and Mental Health in the Pursuit of Sustainable Development Goals"

_nutrients, 2024, doi:10.3390/nu16132036_

Round 1

Reviewer 1 Report

Comments and Suggestions for Authors

1.     This study has dual values of humanitarian care and academic research. It is recommended that the content of the manuscript be slightly adjusted.

2.     For international organizations such as Food and Agriculture Organization (FAO) and the World Health Organization (WHO), except for the full name plus abbreviation for the first time in the manuscript, only the abbreviation may be used thereafter.

3.     This study has few quantitative data or indicators, and lacks description of food types and quantities.

4.     The conclusion of this manuscript is relatively weak and should be presented with quantitative values. In addition, the limitations of this study can also be stated in the conclusion.

Author Response

Reviewer 1 Comments

  1. This study has dual values of humanitarian care and academic research. It is recommended that the content of the manuscript be slightly adjusted.

Response: Thank you very much for the feedback. We understand the reasoning. However, our study encompassing both humanitarian care and academic research on food security and mental health is crucial because it addresses the complex interplay between these issues, fosters comprehensive and effective solutions, and promotes equity and social justice. This integrated approach not only enhances immediate humanitarian responses but also contributes to long-term, sustainable improvements in community well-being.

  1. For international organizations such as Food and Agriculture Organization (FAO) and the World Health Organization (WHO), except for the full name plus abbreviation for the first time in the manuscript, only the abbreviation may be used thereafter.

Response: Thank you. We have rectified this as suggested.

  1. This study has few quantitative data or indicators, and lacks description of food types and quantities.

Response: Thank you. Indeed, quantitative data/indicators and a description of food types and quantities will be an interesting perspective and insight. However, we feel that a twist to this would, to a larger extent, go beyond the specific focus of the current study, which is to investigate the intersection of food insecurity and mental health across various regions and socioeconomic contexts. We choose to prioritize qualitative insights, explore complex social dynamics, and generate nuanced understandings. Indeed, your suggestion is a viable area for further study.

  1. The conclusion of this manuscript is relatively weak and should be presented with quantitative values. In addition, the limitations of this study can also be stated in the conclusion.

Response: Thank you for your feedback and suggestions.  We have included the strengths, areas for improvement and take-home message in the second paragraph of section 8 to strengthen the conclusion. We have also discussed the study's limitations in the last paragraph, highlighted in yellow.

Reviewer 2 Report

Comments and Suggestions for Authors

Helen Onyeaka et al. submitted to Nutrients a review, focusing to the intersection of food security and mental health in the pursuit of SDGs.

This manuscript appears substantially well structured and could be useful for experts in the field.

Here are my suggestions for improvement:

- Detail in the abstract that this research refers to the period 2013-2023;

- Methodology chapter: it is necessary to mention how many papers had been found from the specific databases and how many were finally enrolled from each DB; please provide a detailed table;

- Methodology Chapter: it is essential to explain when the study was conducted (first months of 2024?);

- Please identify the strengths and areas for improvement of your study in the take-home messages.

Comments on the Quality of English Language

Minor editing of English language required

Author Response

Reviewer 2 Comments

Helen Onyeaka et al. submitted to Nutrients a review, focusing to the intersection of food security and mental health in the pursuit of SDGs.

This manuscript appears substantially well structured and could be useful for experts in the field.

Response: We thank you for this constructive comment and appreciate your keen interest in our study.

Here are my suggestions for improvement:

- Detail in the abstract that this research refers to the period 2013-2023;

Response: Thank you for your suggestion. We have included the following statement to reflect your feedback ‘The study involved an analysis of literature search mainly between 2013 and 2023, with an updated addition of relevant 2024 studies’.

- Methodology chapter: it is necessary to mention how many papers had been found from the specific databases and how many were finally enrolled from each DB; please provide a detailed table;

Response: We appreciate the suggestion. However, during the review process, our focus was on ensuring the relevance and quality of the selected articles rather than retaining article counts. As a result, we did not store the specific number of papers found in each database. We prioritised screening and assessing articles for their contribution to the topic rather than tracking the initial search count.

- Methodology Chapter: it is essential to explain when the study was conducted (first months of 2024?);

Response: Thank you for pointing this out. Based on your suggestion, we have added the following statement to section 2.1 of the methodology section, highlighted in yellow: “…with an updated analysis and inclusion of 2024 studies and literature searches up to May 2024”.

- Please identify the strengths and areas for improvement of your study in the take-home messages.

Response: Thank you for your insightful suggestion. We have included the strengths, areas for improvement and take-home message in the second paragraph of section 8.

Again, thank you for allowing us to strengthen our manuscript with your valuable comments, queries, and suggestions. We have worked hard to incorporate your feedback and hope these revisions persuade you to accept our submission.